# Anxiety and depression symptoms among a sample of Khartoum civilians during the 2023 Sudan armed conflict: A cross-sectional study

**Ahmed Balla M. Ahmed**\* , **Ahmed A. Yeddi, Salma S. Alrawa, Esraa S. A. Alfadul**

Faculty of Medicine, University of Khartoum, Khartoum, Sudan

\* m.salahballa@gmail.com

**Data Availability Statement:** All relevant data are within the manuscript and its Supporting Information files.

## Abstract

### Background

Mental health during armed conflicts is of paramount importance, as such situations often lead to increased risks of anxiety and depression symptoms among civilians. The military conflict between the Sudanese army and Rapid Support Forces, which began on April 15, 2023, is currently ongoing mainly in Khartoum State. Despite the significant impact of the conflict on the region, there is a lack of data regarding the mental health status of the residents. The aim of this study is to assess anxiety and depression symptoms among residents of Khartoum State during the first months of the 2023 military conflict.

### Method

We conducted a cross-sectional study among residents of Khartoum State between May 27 and June 19 using an online questionnaire. We used standardized screening questionnaires, namely the Generalized Anxiety Disorder (GAD-7) for anxiety and the Patient Health Questionnaire (PHQ-9) for depression. Multiple logistic regression was used to identify sociodemographic factors that are associated with anxiety and depression symptoms.

### Results

Out of the 393 participants in the study, 70% had symptoms suggestive of depression and 57.3% suffered from anxiety symptoms. Both anxiety and depression were associated with being female (p < 0.001). Being married was a predictor of anxiety (p = 0.028) but not depression (p = 0.3). Other predictors were not significant (p > 0.05).

### Conclusion

High levels of anxiety and depression symptoms were prevalent among Khartoum residents during the conflict, with females and married individuals at higher risk. Immediate medical assessment is essential for identifying cases and providing support. Mental health services should be integrated into emergency response efforts, particularly focusing on vulnerable groups. Future research should address study limitations and explore coping strategies for anxiety and depression in Sudanese adults.

**Funding:** The authors received no specific funding for this work.

**Competing interests:** The authors have declared that no competing interests exist.

## Background

Sudan, a nation situated at the intersection of sub-Saharan Africa and the Middle East, has a history marked by unresolved conflicts and internal strife [1–3]. The country has faced major internal conflicts largely driven by economic, political, and social disparities between the northern and southern regions [4]. These conflicts have resulted in significant loss of life, mass displacements, and famine [4].

The last violent conflict took place on April 15, 2023, between two rival groups in Sudan: the Sudanese Armed Forces (SAF) and the Rapid Support Forces (RSF) [3]. The Sudanese Armed Forces (SAF) is a significant and adequately equipped military force in the nation. It plays a crucial role in addressing internal security and is prepared for potential external threats [5]. The Rapid Support Forces (RSF) is a paramilitary organization established in 2013 to combat insurgent groups in Sudan [5].

The fighting took place mainly in urban centers, which is why civilian casualties were so high [3]. Between April 15 and September 7, approximately 5.1 million people, both inside and outside of Sudan, have been displaced due to the conflict. More than 1 million of them have sought shelter in neighboring countries [6].

People affected by war are at increased risk of poverty, limited access to health care, and food insecurity, in addition to mental health complications including post-traumatic stress disorder (PTSD), anxiety, and depression [7, 8]. Anxiety is a sensation of tension, anxious thoughts, and physical changes like elevated blood pressure. It also includes physical signs like trembling, sweating, disorientation, or an accelerated heartbeat [9]. On the other hand, depression is a prevalent mental illness. It affects 5% of adults worldwide [10]. It involves a range of symptoms, such as low mood, loss of interest and enjoyment, low energy, disturbed sleep and hunger, guilty or low self-worth sentiments, and inadequate attention [10].

All these psychological symptoms are common after disasters and have a major impact on the individuals and communities involved [8]. According to the World Health Organization (WHO), in situations of armed conflict worldwide, 10% of people who experience traumatic events will develop serious mental health problems such as anxiety and depression, and 10% will develop violent behaviors that impede their ability to function effectively [11]. Prior to the current conflict in Sudan, there have been no studies conducted to assess the prevalence of anxiety and depression among the general population. However, it is believed that the country's civil wars have contributed to a rise in mental illnesses like depression, especially among children and women [12].

We aim to study anxiety and depression symptoms as examples of mental health disorders in Khartoum State during the current armed conflict.

## Methods and materials

### Study setting and design

This cross-sectional study was conducted between May 27th and June 19th during the Sudanese armed military conflict of 2023.

### Study population and sample

We targeted male and female residents of Khartoum State who were older than 18 years and were capable of reading and writing in Arabic. Exclusion criteria included substance abuse and psychiatric history. The minimum sample size was calculated using the Cochran formula, following the assumptions of a confidence level (CI) of 95%, an acceptable margin of error of 5%, an expected frequency of 50%, and a target population (N) of unknown. The required

sample size was 384 participants. Given the lack of previous data on the specific prevalence of anxiety and depression during the Sudan armed conflict, a conservative estimate of 50% is used to ensure an adequate sample size for capturing potential trends and variations.

## Data collection tools and procedures

The data were collected using an online self-administered questionnaire that assesses anxiety, depression, and associated factors.

The anxiety was assessed using the instrument of Spitzer et al. (2006) (General Anxiety Disorder: GAD-7) [13]. The scale consists of seven items, asking the respondents how often, during the military conflict, they were bothered by each symptom. According to the GAD-7 developers, a score of 0–4 indicates minimal anxiety, 5–9—mild anxiety, 10–14—moderate anxiety, and ≥15—severe anxiety [13, 14]. The maximum score is 21 points, with a higher score indicating a higher symptom burden. A cutoff of ≥10 marks a probable anxiety diagnosis [15]. The Cronbach alpha in this study was 0.899.

Depression was assessed using the instrument of Kroenke et al., 2001 (Patient Health Questionnaire: PHQ-9) [15]. The PHQ-9 contains nine items. We asked the respondents how often each symptom in the PHQ-9 bothered them during the military conflict. According to the PHQ-9 developers, a score of 0–4 indicates no or minimal depression, 5–9—mild depression, 10–14—moderate depression, 15–19—moderately severe depression, and 20–27—severe depression [16, 17]. A cutoff of ≥10 indicates a probable depression diagnosis [15]. The Cronbach alpha of this study was 0.863. Both instruments (GAD-7 and PHQ-9) were translated from English to Arabic by Pfizer (Pfizer Inc., New York City, NY, USA), and the translated versions were used in this study [18]. A further analysis was conducted on a Saudi sample demonstrating strong internal consistency for the PHQ-9 with a Cronbach's alpha of 0.857, indicating good reliability. Similarly, the GAD-7 showed acceptable reliability with a Cronbach's alpha of 0.763. These findings suggest that the Arabic versions of the PHQ-9 and GAD-7 are valid and reliable tools for screening depression and anxiety [19].

To identify the factors that may be associated with the mental health of the participants, information on demographic characteristics (gender, age, job, where he or she lives, social status, monthly income) was collected.

Due to security issues we couldn't recruit on-ground data collectors, and relied on questionnaire dissemination through social media platforms. In order to improve the diversity and inclusivity of our sample, we utilize various social media platforms to recruit participants. Additionally, we collaborate with influential individuals on social media to expand our outreach and engage with a broader range of community members. For those without social media access, we encourage participants to involve their family and friends in filling out the questionnaire. Prior to enrolling in the study, all participants provided written informed consent. Non-probability sampling was used for data collection.

## Ethics of human subject participation

This study was conducted according to the guidelines laid down in the Declaration of Helsinki, and all procedures involving research study participants were approved by the Health Research Ethics Committee of Gadarif State.

## Data processing and analysis

After extracting data from a Google Form into an Excel sheet, we conducted a comprehensive review to identify and correct inconsistencies, errors, and missing values. Duplicates were eliminated, and participants who did not meet the inclusion criteria were excluded. The data

were coded using numerical values (e.g., assigning 1 for male and 0 for female) and then imported into R software version 4.2.2 for additional analysis. The normality of the distribution was tested using the Shapiro-Wilk test. Descriptive statistics were used for calculating the median and interquartile range for the continuous variables and frequencies with percentages for categorical variables. A multiple logistic regression analysis was performed to identify factors associated with anxiety and depression. Variables with a p-value <0.25 in the univariate logistic regression were included in the multiple logistic regression models. The p-value of $\leq$ .05 was set as the significance level of the study.

## Results

### Demographic characteristics of study participants

A total of 393 participants were included in the study, with a median age of 25 years. The majority of participants were female (n = 288, 73%), and approximately half of the participants were students (n = 196, 50%). Most participants were unmarried (n = 321, 82%) and resided in Khartoum (n = 169, 43%) and Omdurman (n = 136, 35%). A substantial proportion of participants (n = 132, or 34%) reported a monthly income between 33 and 150 thousand "**Table 1**".

**Table 1. Demographic characteristics of the study participants.**

| Variables | N | N = 393[n] | %[IQR] |
|---|---|---|---|
| **Age** | 393 | 25 | (22, 29) |
| **Age group** | 393 | | |
| Youth (18–24 years) | | 194 | 49% |
| Adults (25–64 years) | | 199 | 51% |
| **Gender** | 393 | | |
| Male | | 105 | 27% |
| Female | | 288 | 73% |
| **Occupation** | 393 | | |
| Doctor | | 35 | 8.9% |
| Engineer | | 30 | 7.6% |
| Freelance | | 5 | 1.3% |
| Governmental employee | | 15 | 3.8% |
| Other | | 77 | 20% |
| Pharmacist | | 30 | 7.6% |
| Student | | 196 | 50% |
| Tradesman | | 5 | 1.3% |
| **Residence** | 393 | | |
| Khartoum | | 169 | 43% |
| Khartoum bahri | | 88 | 22% |
| Omdurman | | 136 | 35% |
| **Marital status** | 393 | | |
| Unmarried | | 321 | 82% |
| Married | | 72 | 18% |
| **Monthly income** | 393 | | |
| Less than 33,000 | | 126 | 32% |
| 33,000–150,000 | | 132 | 34% |
| 150,000–250,000 | | 71 | 18% |
| More than 250,000 | | 64 | 16% |

n = Median, IQR = Interquartile Range, N = Target population

## Prevalence of anxiety and depression

The overall median score for the anxiety subscale was 11, with an interquartile range of 9. For the depression subscale, the median score was 14, with an interquartile range of 10. Based on the categorization of participants using GAD-7 and PHQ-9 scores, 31% of participants exhibited severe anxiety, while the majority displayed moderately severe depression (28%). Using a cutoff point of ten, 57.3% of participants were identified as having anxiety symptoms, 73% were identified as having depression symptoms, and 53% were identified as having both anxiety and depression symptoms "Table 2".

## Association between depression and anxiety

A strong positive correlation has been identified between depression and anxiety (correlation coefficient = 0.74, p<0.001) "Fig 1".

## Factors associated with anxiety and depression

Regarding anxiety, gender and marital status emerged as significant predictors. Females had higher odds of experiencing anxiety compared to males (OR = 3.85, 95% CI = 2.37, 6.20, p-value < 0.001). Being married was associated with twice the risk of anxiety compared to being unmarried (OR = 2.20, 95% CI = 1.17, 4.27, p-value = 0.016) "Table 3".

For depression, gender was the only significant predictor. Females had higher odds of experiencing depression compared to males (OR = 3.25, 95% CI = 2.02, 5.23, p-value < 0.001) "Table 4".

## Discussion

This study aims to assess anxiety and depression symptoms as examples of mental health disorders among residents of Khartoum State during the armed conflict that erupted between the Sudanese army and paramilitary forces on April 15, 2023. We found a high prevalence of anxiety and depression among Khartoum residents during the current conflict (57.3% and 73%).

**Table 2. Distribution of participants based on the categorization of their General Anxiety Disorder -7 and Patient Health Questionnaire -9 subscales.**

| Variables | N | N = 393[n] | %[IQR] |
|---|---|---|---|
| **Anxiety N(%)** | 393 | | |
| Mild | | 108 | 27% |
| Minimal | | 60 | 15% |
| Moderate | | 104 | 26% |
| Severe | | 121 | 31% |
| **Depression N(%)** | 393 | | |
| Mild | | 76 | 19% |
| Minimal | | 30 | 7.6% |
| Moderate | | 96 | 24% |
| Moderately severe | | 110 | 28% |
| Severe | | 81 | 21% |
| **GAD-7 score** | 393 | 11 | (6, 15) |
| **PHQ-9 score** | 393 | 14 | (9, 19) |

n = Median, IQR = Interquartile Range, N = Target population, GAD = General Anxiety Disorder, PHQ = Patient Health Questionnaire

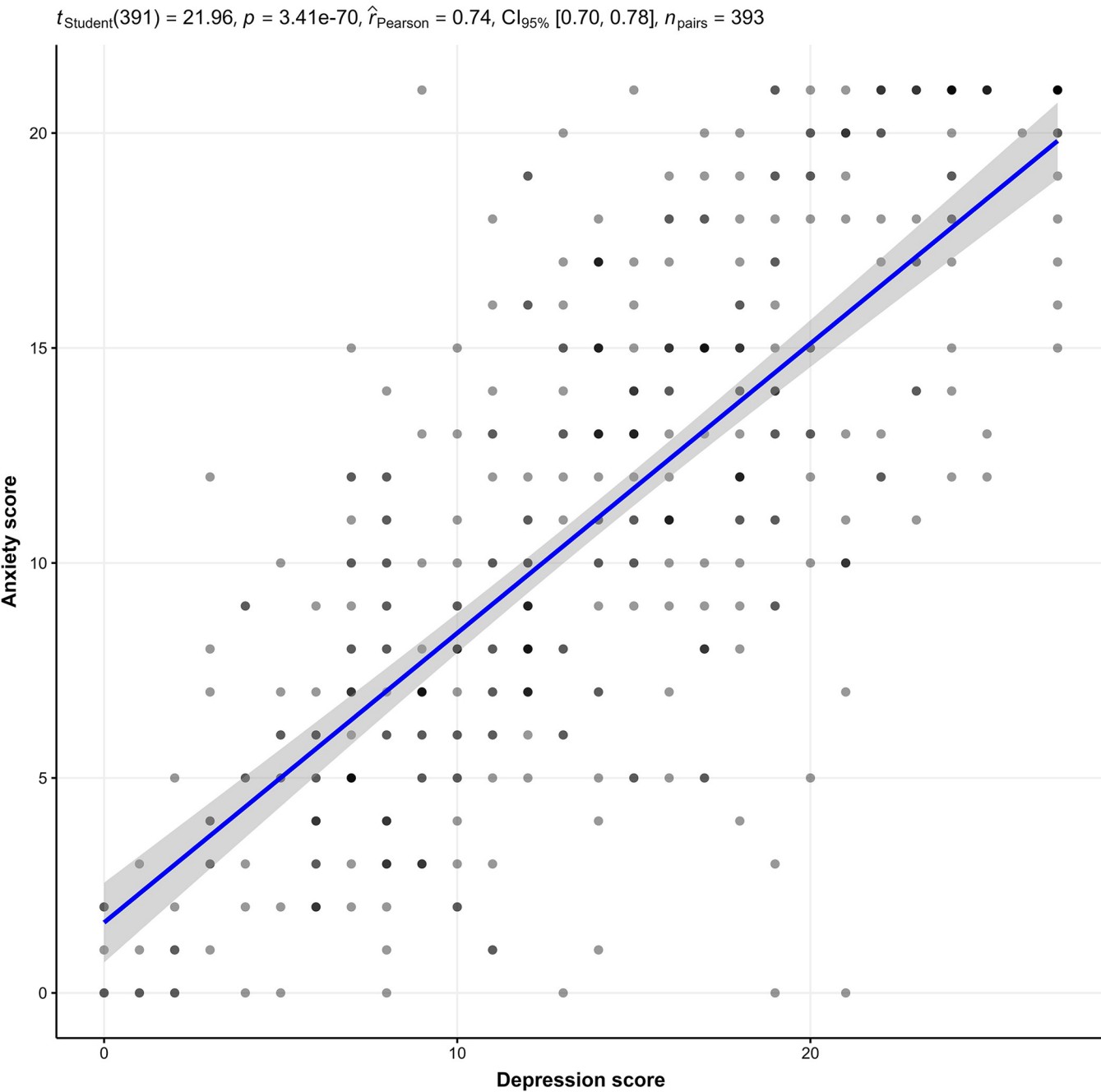

$t_{Student}(391) = 21.96$, $p = 3.41e-70$, $\hat{r}_{Pearson} = 0.74$, $CI_{95\%}$ [0.70, 0.78], $n_{pairs} = 393$

**Fig 1. Association between depression and anxiety.**

The prevalence of both is higher than what is reported among Sudanese cancer patients (26.7% and 41.2%) and internally displaced people in Sudan (23.6% and 24.3%) [20, 21]. The prevalence is also higher than what is reported in Ukraine for both anxiety and depression (35.5% and 42.3%) [22]. One month of war in Ukraine resulted in 294 civilian deaths [23] compared to 604 civilians in Sudan [24]. In both countries, power and water supplies reached a breaking point early in the country [25, 26]. More than half of the participants had symptoms suggestive of both anxiety and depression. A global study revealed that 45.7 of individuals diagnosed

**Table 3. Factors associated with anxiety among the study participants.**

| variables | Univariate | | | Multivariate | | |
|---|---|---|---|---|---|---|
| | OR[1] | 95% CI[1] | p-value | OR[1] | 95% CI[1] | p-value |
| **Gender** | | | | | | |
| Male | — | — | | — | — | |
| Female | 3.66 | 2.30, 5.91 | **<0.001** | 3.80 | 2.37, 6.20 | **<0.001** |
| **Age** | | | | | | |
| Youth | — | — | | — | — | |
| Adults | 1.34 | 0.90, 2.0 | 0.2 | 1.1 | 0.7, 1.74 | 0.7 |
| **Occupation** | | | | | | |
| Doctor | — | — | | | | |
| Engineer | 1.00 | 0.37, 2.73 | >0.9 | | | |
| Freelance | 1.00 | 0.15, 8.33 | >0.9 | | | |
| Governmental employee | 0.76 | 0.22, 2.63 | 0.7 | | | |
| Other | 1.47 | 0.64, 3.38 | 0.4 | | | |
| Pharmacist | 0.76 | 0.28, 2.04 | 0.6 | | | |
| Student | 0.72 | 0.34, 1.49 | 0.4 | | | |
| Trade man | 2.67 | 0.35, 55.1 | 0.4 | | | |
| **Residence** | | | | | | |
| Khartoum | — | — | | | | |
| Khartoum bahri | 1.10 | 0.65, 1.86 | 0.7 | | | |
| Omdurman | 0.86 | 0.55, 1.36 | 0.5 | | | |
| **Marital status** | | | | | | |
| Unmarried | — | — | | — | — | |
| Married | 2.22 | 1.29, 3.97 | **0.005** | 2.20 | 1.17, 4.27 | **0.016** |
| **Monthly income** | | | | | | |
| Less than 33,000 | — | — | | | | |
| 33,000–150,000 | 1.02 | 0.63, 1.67 | >0.9 | | | |
| 150,000–250,000 | 1.35 | 0.75, 2.45 | 0.3 | | | |
| More than 250,000 | 1.29 | 0.70, 2.40 | 0.4 | | | |

[1]OR = Odds Ratio, CI = Confidence Interval

with lifetime major depressive disorder also had a lifetime occurrence of one or more anxiety disorders [27]. Comorbidity is the rule in psychiatric disorders [28].

The prevalence of depression and mental disorders may double in humanitarian crises [29]. High prevalence of mental disorders was reported in Syria [30]. A systematic review in Libya revealed that depression rates could reach 58.6%, and anxiety rates could go up to 56% [31]. Generally, countries in the Middle East have high rates of mental disorders with higher rates in areas of complex emergencies [32]. The current situation in Sudan meets the definition of complex humanitarian emergency [33, 34], and the Sudanese population had a sense of abandonment by the international community as indicated in some reports [35, 36].

It is worth noting that Sudanese people have led a nonviolent revolution in 2019 [37], despite initial success, the triumph was short-lived, and before the second anniversary they were under military rule [37], and as the fourth anniversary a military conflict erupted. We assume that this experience has shattered Sudanese hope of social and political flourishing, and exacerbated their worsened mental distress [38]. Future qualitative research should assess this assumption.

**Table 4. Factors associated with depression among the study participants.**

| variables | Univariate | | | Multivariate | | |
|---|---|---|---|---|---|---|
| | OR[1] | 95% CI[1] | p-value | OR[1] | 95% CI[1] | p-value |
| **Gender** | | | | | | |
| Male | — | — | | — | — | |
| Female | 3.25 | 2.02, 5.23 | **<0.001** | 3.25 | 2.02, 5.23 | **<0.001** |
| **Age** | | | | | | |
| Youth | — | — | | | | |
| Adults | 0.94 | 0.61, 1.46 | 0.8 | | | |
| **Occupation** | | | | | | |
| Doctor | — | — | | | | |
| Engineer | 0.69 | 0.23, 2.03 | 0.5 | | | |
| Freelance | 1.38 | 0.17, 29.0 | 0.8 | | | |
| Governmental employee | 0.69 | 0.19, 2.71 | 0.6 | | | |
| Other | 0.99 | 0.38, 2.41 | >0.9 | | | |
| Pharmacist | 0.95 | 0.31, 2.94 | >0.9 | | | |
| Student | 0.80 | 0.34, 1.76 | 0.6 | | | |
| Tradesman | 1.38 | 0.17, 29.0 | 0.8 | | | |
| **Residence** | | | | | | |
| Khartoum | — | — | | | | |
| Khartoum bahri | 1.09 | 0.61, 1.99 | 0.8 | | | |
| Omdurman | 0.71 | 0.43, 1.16 | 0.2 | | | |
| **Marital Status** | | | | | | |
| Unmarried | — | — | | | | |
| Married | 1.26 | 0.71, 2.31 | 0.4 | | | |
| **Monthly income** | | | | | | |
| Less than 33,000 | — | — | | | | |
| 33,000–150,000 | 1.07 | 0.62, 1.84 | 0.8 | | | |
| 150,000–250,000 | 0.83 | 0.45, 1.58 | 0.6 | | | |
| More than 250,000 | 1.02 | 0.53, 2.02 | >0.9 | | | |

[1]OR = Odds Ratio, CI = Confidence Interval

The military conflict affected the three cities of Khartoum State, namely Khartoum, Omdurman, and Bahri. The city of residence was not a predictor of anxiety and depression levels, as the three of them were roughly similarly affected by the conflict [39].

Regarding the sociodemographic predictors of depression and anxiety symptoms, age was not associated with either of them. Similar findings were reported from a study among Ukrainian combatants [40]. However, studies in the general population showed an association between advanced age and depressive symptoms [41]. One may argue that the war nullifies or reverses the effect of age, as the young adults would worry about their future and lives. Our findings are not conclusive in this regard, as the majority of the participants were in their twenties. There was no association between marital status and depression which is different from findings in Syria and Libya [42, 43]. However, being married was associated with anxiety. A study from Ukraine found that anxiety levels were not related to marital status but to the number of children [41]. We haven't asked about the number of children, yet we speculate that married people are more likely to suffer from anxiety due to concern about their partners and children.

There was a significant association between being female and having both anxiety and depressive symptoms. This is consistent with findings from the literature, as females are

vulnerable to anxiety and depression [44, 45]. Financial stress is recognized as a determinant of depression [46, 47], but no association was found in our study. During the current conflict, the capital became a lawless region, and looting became the norm. Even non-governmental organizations were not immune from the attacks [48, 49]. Wealthy families are not immune; their houses, businesses, and possessions are at risk of being shelled or stolen. The poor, on the other hand, can't afford the prices that skyrocketed during the conflict [50]. This may explain why financial status was not a determinant of depression.

## Strengths and limitations

This is the first study to assess the anxiety and depression symptoms levels among Khartoum residents during the current military conflict. It is also one of the few studies to assess mental health early during a military conflict in the Middle East. This documentation may guide interventions and serve as a reference in post-conflict assessment.

The findings of this study, however, should be viewed in light of the convenience sampling limitation. The access to the internet is a double edged sword as it can be used to access materials which may intensify or diminish the mental distress. We were forced to use this sampling technique due to accessibility issues caused by disturbed web connections; thus, the results of this study are not generalizable to the public. The cross-sectional nature of the study is another limitation and adopting a longitudinal design would be more appropriate for establishing temporal or causal relationships. While demographic variables were outlined, important conflict exposure indicators like experiencing abuse were not included. Both GAD-7 and PHQ-9 are screening tools; psychiatric interviews are needed to estimate the actual prevalence of anxiety and depression.

## Conclusion

We found a high prevalence of anxiety and depression symptoms among Khartoum residents during the first two months of the military conflict. Females were more vulnerable to anxiety and depression, and married individuals were more susceptible to anxiety. The high levels of anxiety and depression symptoms are alarming and warrant medical attention to confirm the cases and offer the needed support. As was suggested in Ukraine, mental health should be part of emergency and recovery solutions [51]. The groups which are more likely to experience anxiety and depression, like females, should be prioritized. Future studies should aim to overcome the limitations of this study. It is essential to explore the coping mechanisms employed by Sudanese adults to manage anxiety and depression, evaluating their efficacy. Additionally, we should investigate other factors that could intensify mental distress, such as escalating unemployment and inflation.

## Supporting information

**S1 Data. Dataset for the entire study.**
(XLSX)

## Acknowledgments

We are exceedingly indebted to all participants who made time to participate in this study. The study could not have been accomplished without their participation.

## Author Contributions

**Conceptualization:** Ahmed Balla M. Ahmed, Ahmed A. Yeddi.

**Data curation:** Ahmed Balla M. Ahmed.

**Formal analysis:** Esraa S. A. Alfadul.

**Investigation:** Salma S. Alrawa.

**Methodology:** Ahmed Balla M. Ahmed, Ahmed A. Yeddi, Salma S. Alrawa, Esraa S. A. Alfadul.

**Project administration:** Ahmed Balla M. Ahmed.

**Resources:** Ahmed Balla M. Ahmed, Ahmed A. Yeddi.

**Writing – original draft:** Ahmed Balla M. Ahmed, Ahmed A. Yeddi, Salma S. Alrawa, Esraa S. A. Alfadul.

**Writing – review & editing:** Ahmed Balla M. Ahmed, Ahmed A. Yeddi, Salma S. Alrawa, Esraa S. A. Alfadul.

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
