## [Decision Letter · Decision Letter 0]

22 Jan 2024

PONE-D-23-30028Anxiety and depression among a sample of Khartoum civilians during the 2023 Sudan armed conflict: A cross-sectional studyPLOS ONE

Dear Dr. M. Ahmed,

Thank you for submitting your manuscript to PLOS ONE. After careful consideration, we feel that it has merit but does not fully meet PLOS ONE’s publication criteria as it currently stands. Therefore, we invite you to submit a revised version of the manuscript that addresses the points raised during the review process.

**ACADEMIC EDITOR: **After carefully reviewing your work, we advise that you make the necessary changes, paying particular attention to the alignment, line spacing, syntax, results, and discussion domains, as suggested by the comments below. Kindly submit the updated document following Plos One Publication's standards. best wishes!

We look forward to receiving your revised manuscript.

Kind regards,

Yadeta Alemayehu

Academic Editor

PLOS ONE

Journal Requirements:

Additional Editor Comments:

Please include the following in the revised version:

1. On Ethics statement please include what you have done for those peoples you encountered with severe depression or anxiety while you are doing the study.

2. Please align with the publication guideline of Plos One i.e focus on alignment, line spacing guideline.

3. its better to describe some finding by using graphical expressions like bar graph or pie chart if appropriate.

4. Some contents of the table were left empty. However, every value you get from your study is significant if reported in number.

5. Some notes you wrote under the table seems senseless. Its not clear why you wrote them under the table unless they are represented by symbol in the table.

6. Some reasons you put under the discussion need evidence or reference

Reviewers' comments:

Reviewer's Responses to Questions

**Comments to the Author**

1. Is the manuscript technically sound, and do the data support the conclusions?

Reviewer #1: Partly

Reviewer #2: Yes

Reviewer #3: Yes

2. Has the statistical analysis been performed appropriately and rigorously? 

Reviewer #1: No

Reviewer #2: Yes

Reviewer #3: Yes

3. Have the authors made all data underlying the findings in their manuscript fully available?

Reviewer #1: Yes

Reviewer #2: No

Reviewer #3: Yes

4. Is the manuscript presented in an intelligible fashion and written in standard English?

Reviewer #1: Yes

Reviewer #2: Yes

Reviewer #3: Yes

5. Review Comments to the Author

Reviewer #1: Dear Researcher! Thank you for conducting this study. Your manuscript has no line numbers to comment on by line number and page. The method and materials section of this manuscript was not clearly written and understandable.

Reviewer #2: This research was conducted to understand the mental health status of the inhabitants of Khartoum State amidst the armed conflict that broke out on April 15, 2023. Overall, the manuscript is well-written and provides a good overview of the study problem. However, it could be improved by addressing the following points:-

-Abstract: begin with that sentence about importance of mental health during armed conflicts, then move to the current situation in Khartoum. Specify the sample size used in the study. Suggest specific recommendations for mental health interventions that could be implemented in Khartoum state to address the high prevalence of anxiety and depression.

-The introduction provided some unnecessary background information on Sudan's history (reasons of the war in south Sudan in page 3 and details about the two warring parties in page 4). The introduction overly focuses on historical background in Sudan. This may detracts from the main focus of the research, which appears to be the study of anxiety and depressive disorders during the current armed conflict. The authors need to condense the historical background information and provide a concise overview that is directly relevant to the research topic.

-Introduction: regarding this sentence " Sudan, one of the countries in the Middle East, is located at the crossroads of sub-Saharan Africa and the Middle East,", the WHO divides the world into six regions and Sudan is classified as an Eastern Mediterranean country. However, If the authors preferred to describe the geography, it would be better to say (a north African country) or (a nation situated at the intersection of sub-Saharan Africa and the Middle East).

-Introduction: this paragraph needs paraphrasing "WHO has estimated that in situations of armed conflict worldwide, 10% of people who undergo traumatic events will experience serious mental health problems ….etc" . I suggest: "According to the World Health Organization (WHO), in situations of armed conflict worldwide, 10% of people who experience traumatic events will develop serious mental health problems ….etc"

-Methods: this sentence is not clear " The research complies with the ethical standards for conducting this study, as indicated by Gadarif state, Ministry of Health ethical committee. " what is the relevance of Gadarif state here? Also, the information about "informed consent" should be written in this section

-Methods: The use of an online self-administered questionnaire through Google Forms is appropriate and convenient during a conflict situation. However, it would be helpful to discuss any measures taken to ensure the representativeness of the sample

-Methods: Regarding sample size, it would be beneficial to provide a rationale or reference supporting the assumption of an expected frequency of 50%

-Methods: it would be helpful to provide further information on this Arabic form and its validity of if available. Are there any differences or modifications made on that Arabic form?

-Discussion: The discussion infers that the recent memories of revolutions in the Middle East countries led to high levels of anxiety and depression in Sudan !?. I do not think that this is a reasonable or convincing hypothesis.

-Discussion: the authors could discuss what are the implications of the findings for mental health interventions (i.e. How can the findings be used to improve mental health services for people affected by armed conflict) and Suggest directions for future research.

Limitations: There are some limitations which are not mentioned; cross-sectional design precludes inferences about temporal or causal relationships between conflict exposure and mental health outcomes. A longitudinal cohort study design would better characterize changes over time. Also, Convenience sampling via social media limits generalizability, as those without internet access are excluded. Lastly, Demographic variables are described but important conflict exposure indicators weren’t included (e.g. violence witnessed, injury, displacement).

Reviewer #3: Dear authors

The manuscript "Anxiety and depression among a sample of Khartoum civilians during the 2023 Sudan armed conflict: A cross-sectional study" aimed "to assess anxiety and depression among residents of Khartoum state during the first months of the 2023 military conflict"

The abstract is structured, clear and appropriate.

The introduction is clear and adequate. Justifies the relevance of the study. Defines the concepts to be operationalized and the context of the study. Adequate objective.

The method is adequately well described. It justifies the methodological options. Presents a sample with inclusion and exclusion criteria. Describes the variables and statistical analyses. It does not mention whether the study was approved by an ethics committee, only that it followed ethical precepts in accordance with the country's legislation.

Results presented under tables. They are clear and adequate.

Appropriate discussion. The findings were compared with previous studies carried out in a conflict context. It presents limitations and practical implications. Clear and adequate conclusion.

Best regards

6. PLOS authors have the option to publish the peer review history of their article (what does this mean?). If published, this will include your full peer review and any attached files.

Reviewer #1: No

Reviewer #2: No

Reviewer #3: **Yes: **Luís Sousa

---

## [Author Response · Author response to Decision Letter 0]

12 Mar 2024

Editor comments:

-Please ensure that your manuscript meets PLOS ONE's style requirements, including those for file naming

I modified it according to your recommendations.

-Please include captions for your Supporting Information files at the end of your manuscript, and update any in-text citations to match accordingly.

We didn't upload any supporting Information file.

-On Ethics statements please include what you have done for those peoples you encountered with severe depression or anxiety while you are doing the study.

Due to the anonymous nature of our questionnaire for ethical reasons, we were unable to immediately pinpoint specific individuals with severe depression or anxiety who took part in the survey. Nonetheless, mental health support initiatives by psychiatrists were shared on the same social media platforms we utilized.

-Please align with the publication guideline of Plos One i.e focus on alignment, line spacing guideline

We revised the manuscript and modified it according to your publication guidelines.

-its better to describe some finding by using graphical expressions like bar graph or pie chart if appropriate

Your feedback is valuable. We represented the correlation between anxiety and depression using a graph.

-Some contents of the table were left empty. However, every value you get from your study is significant if reported in number

We revised all the tables and modified them accordingly.

-Some notes you wrote under the table seem senseless. It's not clear why you wrote them under the table unless they are represented by symbols in the table.

We modified it and put the symbols in the table then clarified them under the tables.

-Some reasons you put under the discussion need evidence or reference.

We modified the discussion accordingly. Thank you for your input.

Reviewer comments:

Reviewer #1:

Abstract:

- It contains all the scientific content and is written well.

Thank you for the comment.

Introduction:

- Introduction correct as “Background”

We changed it.

- Background: It is more focused on the history of Sudanese politics and war history than common mental disorders during and after military conflicts, and it is better to focus on and rewrite more facts and gaps about common mental illnesses that need more research and more studies on anxiety and depression disorders among people living in conflict or war zones (Khartoum State)

We condensed the historical background and added more facts about mental illnesses in Sudan.

Methods:

- Materials and methods can be correct as "Methods and materials."

We changed it.

- For this cross-sectional study, study setting, design, period, source of population, sample size determination and sampling technique, data collection tools and procedures, operational definition, and data processing and analysis were not separately and clearly written. So, it is better if all the above subtopics under Methods and Materials are rewritten in a brief and clear manner. 

We made the suggested modifications. It really helped clarify this section. Thanks for your comment.

- It is not clear how study participants were selected from the study population. Did you use probability or non-probability techniques? The authors were expected to state the procedures they had used to select quantitative study participants.

We used a non-probability sampling technique and clarified this in this version of the manuscript. 

- You have stated that “this prospective cross-sectional study was conducted between May 27 and June 19, 2023, during the armed military conflict in Khartoum State, the capital of Sudan.” A prospective study design is quite different from a cross-sectional study. The authors should explain what a prospective cross-sectional study means

We corrected it to be (a cross-sectional study) without prospective which doesn't fit the design as stated by the reviewer. 

- As exclusion criteria, the authors stated that "substance abuse and psychiatric history”. How did you differentiate or screen those individuals who had been abusing substances and living with a psychiatric history? Or, what kind of procedures had been used to screen those people who had been abused by substances and had psychiatric disorders?

We asked the participants at the start of the questionnaire if they had a psychiatric history (diagnosed by a psychiatrist) or abused any substance. Those acknowledging substance abuse or a psychiatric history were excluded from the study. We believe the anonymous nature of the questionnaire encourages self-identification as it eliminates potential stigma associated with disclosure.

- You stated that “data were collected using an online self-administered questionnaire through Google Forms due to their publicity." How were study participants selected? What kind of sampling technique had been used?

We used a non-probability sampling technique and clarified this in this version of the manuscript. 

- Data were collected using an online self-administered questionnaire through Google Forms. However, you stated that “we asked respondents" or asking respondents—it seems like a face-to-face interview. Please try to rewrite it clearly.

We paraphrased it.

- For this study, the English version of the questionnaire was translated to the Arabic version, but its contents were not well addressed.

We used the formally translated version as stated in data collection tools and procedures: "Both instruments (GAD-7 and PHQ-9) were translated from English to Arabic by Pfizer (Pfizer Inc., New York City, NY, USA), and the translated versions were used in this study [18]. A further analysis was conducted on a Saudi sample demonstrating strong internal consistency for the PHQ-9 with a Cronbach's alpha of 0.857, indicating good reliability. Similarly, the GAD-7 showed acceptable reliability with a Cronbach's alpha of 0.763. These findings suggest that the Arabic versions of the PHQ-9 and GAD-7 are valid and reliable tools for screening depression and anxiety [19]".

- The Statistical analysis plan can be corrected to “Statistical analysis.”

We changed it into "Data processing and analysis" as recommended in a previous comment. 

- It is not clear how the data were cleaned, coded, and entered into an Excel sheet. 

We clarified it. "After extracting data from a Google Form into an Excel sheet, we conducted a comprehensive review to identify and correct inconsistencies, errors, and missing values. Duplicates were eliminated, and participants who did not meet the inclusion criteria were excluded. The data were coded using numerical values (e.g., assigning 1 for male and 0 for female) and then imported into R software version 4.2.2 for additional analysis". Thank you for this comment.

Results:

- Demographic Characteristics of Study Participants correct it as” Demographic characteristics of study participants”

We changed it.

- A total of 393 participants were included in this study. However, within the Methods and Materials section, the sample size was 384; these two figures are a little different from each other. 

The minimum required sample was 384, so the included participants met the sample size and increased from it. 

- The total number of single study participants were 314 (80%), which implies that being single is a risk factor for developing common mental illnesses (anxiety and depression). So why were the majority of your study participants single?

Relying solely on social media platforms may boost the involvement of unmarried individuals in the survey. It is evident that older participants, who are married, may face challenges with technology literacy, particularly in utilizing social media in Sudan.

We believed that this was one of the limitations and we addressed it in the limitation section and clarified that the current war led us to use this way of data collection.

- Table 1, Standard age classification was not done 

We added the standard age classification as recommended.

-Residence: Khartoum State was classified into three sub-cities: Khartoum, Khartoum Bahri, and Omdurman. What is the significance of classifying Khartoum state into sub cities for this study?

Khartoum State geographically is divided into three main cities: Khartoum, Bahri, and Omdurman.

We asked participants in which city they live to determine if their place of residence had an impact on their anxiety and depression symptoms. Our analysis revealed that residency did not have any influence on anxiety and depression symptoms.

- Marital status: – total number of divorced = 6(1.5%)

 -total number of widowed 1(0.3) 

I think model fitness was not checked for these two variables (divorced and widowed), so try to run the row data again to check its model fitness and merge those less than 5%. 

We combined the categories of single, divorced, and widowed into a single group labeled as unmarried.

Prevalence of Anxiety and Depression:

- Among study participants, 31% of them exhibit severe anxiety, 28% of them have moderate-to-severe anxiety, and using the cutoff point, 57.3% of study participants were anxiety patients. Similarly, 73% of study participants had depression. What about those study participants who had both anxiety and depression (comorbidity between anxiety and depression)? OR, the prevalence of those studies participants who had met criteria for both anxiety and depression during study period? Similarly, what about other common anxiety disorders, such as PTSD and panic disorder, which are common during and after military conflict?

We clarified that "53% were identified as having both anxiety and depression symptoms" in the result section.

Our study specifically focused on measuring anxiety and depression symptoms, and did not assess other mental disorders. We anticipated that our future studies will investigate these additional mental health conditions.

- What is the association between anxiety and depression, according to this study?

We clarified it in the result section "A strong positive correlation has been identified between depression and anxiety (correlation coefficient =0.74, p<0.001)".

 Factors Associated with Anxiety and Depression:

- Factors Associated with Anxiety and Depression, can be corrected as “Factors associated with anxiety and depression”

We changed it.

- Being married was associated with anxiety. (Being married was associated with twice the risk of anxiety compared to being single (OR = 2.21, 95% CI = 1.11, 4.57, p-value = 0.028). But most literature reports that being married is a protective factor for mental illnesses, including anxiety. The authors expected to justify with scientific evidence why being married is associated with anxiety.

We discussed it in the discussion section. 

- Table 3 and table 4 no standard age classification is done 

We added it as recommended.

Discussion:

- You stated that your study aims “to document the mental status of residents of Khartoum State during the armed conflict that erupted between the Sudanese army and paramilitary forces on April 15, 2023”. But the mental status of an individual is different from mental illness (mental disorder) in many ways. The mental status of a person is a one-time clinical examination finding that is not similar to or equal to a mental disorder. Mental status clinical findings might change from time to time; however, mental disorders (anxiety and depression) are syndromes that can be characterized by behavioral, emotional and autonomic symptoms and sustain for a long period of time. For instance, general anxiety disorder is not diagnosed based on a month's duration of symptoms; at least six consecutive months of experiencing anxiety symptoms by the person are mandatory.

We thank the reviewer for his thoroughness in explaining this matter. We made changes accordingly.

- Your discussion is too shallow to address the discrepancies between previous study findings and your study findings.

We modified the discussion to address these discrepancies in more detail.

- You have stated that “Sudanese have witnessed the consequences of military conflict in Libya, Syria, and Yemen. This drove them to lead a non-violent revolution in 2019 as they aimed to have democracy”. Sudanese were not physically witnessing the consequences of conflict, similar to other nations; instead, they had heard and watched the conflict through different media. So what distinguishes Sudanese people from other neighboring countries? According to this study, the exaggerated prevalence of anxiety (57.3%) and depression (73%) was not well justified with scientifically sound

We deleted this part of the discussion and updated the discussion of possible reasons for this high prevalence.

Strengths and limitations:

- Well done, but the main limitation of this study was not stated well. 

We updated this section.

Conclusion:

- Your recommendations were not in line with your study findings. For example, your study did not assess and identify mental health care gaps, and lack of emergency mental health services in Khartoum State. 

Though we haven't particularly addressed the gaps in mental healthcare , we believe the high levels of anxiety and depression warrants attention. Nonetheless, we kept it brief in this version and added other recommendations based on our findings.

Other points:

- I have a concern regarding your study title “Anxiety and depression among a sample of Khartoum civilians during the 2023 Sudan armed conflict: A cross-sectional study”. Because normal anxiety and depressive symptoms are common for people living in war zones, Normal anxiety symptoms are quite different from those of pathological anxiety disorder in the following parameters:

Intensity: -in cases of pathological anxiety the person experiences relatively high and/or out of proportion to the situation or circumstances, and in cases of normal anxiety relatively low and/or proportionate to the situation or circumstances.

Duration:- In cases of pathological anxiety, the person generally experiences longer-lasting or recurrent symptoms. But, in the case of normal anxiety, it is generally shorter-lasting.

Preoccupation: - with anxiety is common among patients with pathological anxiety and not common among those with normal anxiety 

Effects on behavior and functioning: - Pathological anxiety causes long-standing changes in behavior and impairs functioning. But, generally, it does not affect behavior more than temporarily and does not impair functioning.

Quality of the experience: - Pathological anxiety is distressing, overwhelming, and incapacitating. However, normal anxiety is unpleasant, but not too disturbing or distressing for a long time. So it is too difficult to differentiate normal anxiety from pathological anxiety by analyzing the collected online data.

It is better to modify your study title to “Anxiety and depression symptoms among a sample of Khartoum civilians during the 2023 Sudan armed conflict: A cross-sectional study” . Because of the above factors, your study results were highly exaggerated. “Among the study participants, 70% had depression, and 57.3% suffered from anxiety."

Your advice is really appreciated, we changed the title according to your recommendation. We acknowledged certain limitations that may have influenced the presentation of the results, which we have outlined in the limitations section of our research.

Reviewer #2:

- This research was conducted to understand the mental health status of the inhabitants of Khartoum State amidst the armed conflict that broke out on April 15, 2023. Overall, the manuscript is well-written and provides a good overview of the study problem. 

Thank you very much for this comment.

Abstract:

- begin with that sentence about the importance of mental health during armed conflicts, then move to the current situation in Khartoum. Specify the sample size used in the study. Suggest specific recommendations for mental health interventions that could be implemented in Khartoum state to address the high prevalence of anxiety and depression

We changed it accordingly. Thank you for this comment.

Introduction:

- The introduction provided some unnecessary background information on Sudan's history (reasons for the war in South Sudan in page 3 and details abo

---

## [Decision Letter · Decision Letter 1]

10 Jul 2024

Anxiety and depression symptoms among a sample of Khartoum civilians during the 2023 Sudan armed conflict: A cross-sectional study

PONE-D-23-30028R1

Dear Dr. Ahmed,

We’re pleased to inform you that your manuscript has been judged scientifically suitable for publication and will be formally accepted for publication once it meets all outstanding technical requirements.

Kind regards,

Yadeta Alemayehu

Academic Editor

PLOS ONE

Reviewers' comments:

Reviewer's Responses to Questions

**Comments to the Author**

1. If the authors have adequately addressed your comments raised in a previous round of review and you feel that this manuscript is now acceptable for publication, you may indicate that here to bypass the “Comments to the Author” section, enter your conflict of interest statement in the “Confidential to Editor” section, and submit your "Accept" recommendation.

Reviewer #4: All comments have been addressed

Reviewer #5: All comments have been addressed

2. Is the manuscript technically sound, and do the data support the conclusions?

Reviewer #4: Yes

Reviewer #5: Yes

3. Has the statistical analysis been performed appropriately and rigorously? 

Reviewer #4: Yes

Reviewer #5: Yes

4. Have the authors made all data underlying the findings in their manuscript fully available?

Reviewer #4: Yes

Reviewer #5: Yes

5. Is the manuscript presented in an intelligible fashion and written in standard English?

Reviewer #4: Yes

Reviewer #5: Yes

6. Review Comments to the Author

Reviewer #4: Appreciate to the author for incorporating all the revisions based on the reviewer's comments and providing explanations for each change. The revised manuscript demonstrates improved design and writing, with a clear and cohesive introduction and discussion section. Overall, the revised manuscript meets the criteria for acceptance.

Reviewer #5: (No Response)

7. PLOS authors have the option to publish the peer review history of their article (what does this mean?). If published, this will include your full peer review and any attached files.

Reviewer #4: No

Reviewer #5: **Yes: **Ayesha Ahmad

---

## [Editor Report · Acceptance letter]

16 Jul 2024

PONE-D-23-30028R1 

PLOS ONE

Dear Dr. M. Ahmed, 

I'm pleased to inform you that your manuscript has been deemed suitable for publication in PLOS ONE. Congratulations! Your manuscript is now being handed over to our production team.

Kind regards, 

on behalf of

Yadeta Alemayehu 

Academic Editor

PLOS ONE